# An Arsenite Relay between PSMD14 and AIRAP Enables Revival of Proteasomal DUB Activity

**DOI:** 10.3390/biom11091317

**Published:** 2021-09-06

**Authors:** Sigalit Sukenik, Ilana Braunstein, Ariel Stanhill

**Affiliations:** 1Department of Biochemistry, School of Medicine, Technion, Haifa 31096, Israel; sigalit4f@gmail.com (S.S.); bilana@tx.technion.ac.il (I.B.); 2Department of Natural and Life Sciences, Open University of Israel, Ra’anana 43710, Israel

**Keywords:** AIRAP, PSMD14/RPN11, proteasome, protein misfolding, proteostasis, arsenite

## Abstract

Maintaining 26S proteasome activity under diverse physiological conditions is a fundamental requirement in order to maintain cellular proteostasis. Several quantitative and qualitative mechanisms have evolved to ensure that ubiquitin–proteasome system (UPS) substrates do not accumulate and lead to promiscuous protein–protein interactions that, in turn, lead to cellular malfunction. In this report, we demonstrate that Arsenite Inducible Regulatory Particle-Associate Protein (AIRAP), previously reported as a proteasomal adaptor required for maintaining proteasomal flux during arsenite exposure, can directly bind arsenite molecules. We further show that arsenite inhibits Psmd14/Rpn11 metalloprotease deubiquitination activity by substituting zinc binding to the MPN/JAMM domain. The proteasomal adaptor AIRAP is able to directly relieve PSMD14/Rpn11 inhibition. A possible metal relay between arsenylated PSMD14/Rpn11 and AIRAP may serve as a cellular mechanism that senses proteasomal inhibition to restore Psmd14/Rpn11 activity.

## 1. Introduction

Maintaining proteome functionality is a major challenge that the cell must resolve in order to survive the diverse conditions that it encounters. Changes in biophysical conditions and post-translational modifications all impact the correct folding of proteins, and cellular machineries have evolved to maintain proteome integrity. Promiscuous interactions of misfolded proteins are highly toxic to the cell and manifest in various pathologies [1]; therefore, restoring the proper configuration of misfolded proteins by chaperones, reducing enzymes and more constitutes a salvation pathway. In contrast, degradation pathways such as the ubiquitin–proteasome system (UPS) or the lysosomal pathway eliminate the toxicity of misfolded proteins by proteolytic degradation [2]. The 26S proteasome is the central module of the UPS to which all designated proteins, usually ubiquitinated with suitable configurations of ubiquitin, are delivered. The 26S proteasome is composed of a 20S catalytic particle and a 19S regulatory particle that enables recognition, deubiquitination and unfolding of a UPS substrate. In order to ensure the functionality of the basic 26S proteasome module, various modes of regulation exist in order to maintain functionality during acute and chronic conditions. We can artificially divide these modifications into several categories, including quantitative and qualitative changes in proteasome subunits (executed by transcriptional [3,4,5] and translational [6] programs as well as post-translational modifications of proteasome subunits [7]) and changes in the composition of specific subunits, as exemplified in the case of the immunoproteasome (20Si) and the thymus proteasome (20St). In this mode of regulation, the exchange of catalytic subunits in response to acute signaling (IFN-γ) [8,9] or unique surroundings [10,11] enables modification of the proteasome to a more suitable peptide output. Another mode of regulation consists of alternating the entire cap composition. Alternative caps such as PA200 enable the engagement of the proteasome with specific substrates [12,13], while the PA28 (11S) enables proteasome modulation during specific conditions such as IFN-γ [9,14,15]. The fourth mode of regulation that is diverse and continuously engaged with the proteasome involves shuttling factors. These accessory factors typically consist of a proteasome binding domain that usually consists of a ubiquitin-like (UBL) domain that binds one of the proteasome ubiquitin receptors and several ubiquitin binding domains (UBD) that account for ubiquitin chain affinity and specificity (for reviews, see [16,17,18]). The UBA–UBL mode of substrate delivery to the proteasome increases the diversity and flux of UPS substrates [19], induces proteasome activation [20] and supports the formation of liquid–liquid phase separation of proteasomes [21,22].

Originally identified as an arsenite inducible gene [23], AIRAP/Zfand2a was subsequently found to bind proteasomes with unique biochemical characteristics [24]. These include AIRAP–proteasome integrity in the absence of ATP and enhanced peptide hydrolysis. Unlike other proteasomal adaptors, AIRAP–proteasome binding is not obtained via UBA domains, nor does it bind ubiquitin. AIRAP knockout cells accumulate higher polyubiquitin levels in response to arsenite exposure (typical of UPS impairment) and proteasomes acquired from the knockout cells contain higher levels of polyubiquitinated substrates [24]. These findings rule out AIRAP’s function as a shuttle factor but rather indicate it as a proteasome adaptor required for the efficient processing of UPS substrates by the proteasome upon arsenite-inflicted inhibition. In line with this notion, AIRAP expression was found to be induced upon proteasomal inhibition [25]; thus, AIRAP’s role and gene expression programing point towards a mechanistic function in retaining proteasomal function upon inhibitory conditions, yet the mechanistic details of AIRAP function have not been revealed. In this report, we shed light on the mechanistic role of AIRAP in proteasome regulation and identify AIRAP as a metal-binding protein that regulates the deubiquitination activity of Psmd14/Rpn11.

## 2. Materials and Methods

### 2.1. Cell Transfection, Lysis and Protein Purification

HEK293 and MEF cell lines were cultured in complete medium (DMEM supplemented with 10% heat-inactivated fetal bovine serum, glutamine, non-essential amino acids, penicillin–streptomycin and 55 μM 2-mercaptoethanol). Cell lysis was performed in TNH buffer (20 mM HEPES pH 7.9, 100 mM NaCl, 1% Triton X-100, 1 mM EDTA, 1.5 mM MgCl_2_, 1 mM DTT and protease inhibitors). Lysates were centrifuged at 17,000× *g*, 4 °C, for 10 min and supernatants were used for further analysis. In 26S proteasome purifications, ATP (4 mM) was added to the TNH buffer in order to preserve the proteasome complex. In AIRAP proteasome purifications, hexokinase (135 µg/mL) with glucose (20 mM) was added to the TNH buffer in order to deplete ATP required for AIRAP–proteasome integrity [24].

Then, 26S proteasome purifications from lysate were performed by immunoprecipitation, as previously described using PSMA1 Mab 2–17 hybridoma [26] or S5a CPTC-PSMD4 hybridoma [27]. Where applicable, soluble 26S proteasomes or AIRAP-containing proteasomes were eluted from beads by incubation with a 20S PSMA1 peptide (100 μg/mL; sequence: Ac-KAQPAQPADEPAE-NH_2_), S5a peptide (200 μg/mL; sequence: NNEAIRNAMGSLASQATKDGKKDKKEEDKK) or a GST-S5a protein (150 μg/mL), 850 RPM, for 1 h, R.T.

For evaluation of Psmd14 MPN/JAMM motif requirement for PAO binding, cells were transfected with mammalian expression vectors expressing YFP-tagged PSMD14, Flag-Psmd14 Mut (pCDNA3.1) and Flag-tag GFP, used as a specificity control. Nucleotide mutations for Flag-Psmd14 (His113/115 Ala) were performed using Quick Change Site-Directed Mutagenesis (Stratagene). All constructs were verified by sequencing. Transient transfections were performed using the calcium phosphate precipitation method and cell lysates were evaluated 36–48 h post-transfection.

### 2.2. Proteasomal Psmd14 DUB Assay

Proteasomes containing only Psmd14 and not Uch37 or Usp14 DUBs were prepared as previously described [28]. Briefly, lysis of HEK293 cells was performed in low-salt buffer (10 mM NaCl, 75 mM phosphate buffer pH 7.5, 10% glycerol, 1 mM EDTA, 5 mM MgCl_2_, 0.5% NP40, 1 mM DTT and protease inhibitors). S5a purified proteasomes were then washed with high-salt buffer (750 mM NaCl, 75 mM phosphate buffer pH 7.5, 10% glycerol, 1 mM EDTA, 5 mM MgCl_2_, 0.5% NP40, 1 mM DTT), twice for 10 min. After incubation, proteasomes were restored back to low-salt buffer. Proteasomal deubiquitination was performed by incubating the purified 19SΔUU in the presence of polyubiquitinated Sic1 and samples were withdrawn into Laemmli buffer at the indicated time points. Proteasomes were untreated or arsenite-treated (4 uM) with or without recombinant AIRAP as indicated. Then, 26S vs. 19SΔUU DUB content was evaluated by Psmd14, Uch37 and Usp14 immunoblots. AIRAP, Psmd14 and Sic1 content was evaluated by immunoblot.

### 2.3. Sic1 Ubiquitination

For the in vitro ubiquitination of Sic1, the following proteins were produced: E1 was produced as previously described [29], the UBC4 E2, RSP5 E3 and T7 tagged Sic1 expression plasmids (a kind gift from Y. Saeki) were produced as previously described [30]. Ubiquitin (U6253, Sigma-Aldrich: St. Louis, MO, USA), E1, UBC4 E2, RSP5 E3 and T7 tagged Sic1 were incubated in activity buffer (50 mM Tris pH 7.5, 2 mM ATP, 10 mM creatine phosphate, 0.1 mg/mL creatine phosphate kinase, 0.1 mM β-mercaptoethanol and 5 mM MgCl_2_) for 2 h at RT. The reaction was resuspended in 500 μL PBS buffer and T7 tagged Sic1 WT was Ni-NTA affinity purified and imidazole eluted. Sic1 content was revealed by T7 immunoblots.

### 2.4. Antibodies

Antisera to AIRAP has been previously described [23]. Polyclonal Psmd14 antiserum was produced in rabbit by immunization with recombinant full-length mouse Psmd14 protein. The sources for the following antibodies were: Anti-Ubiquitin (Zymed 131600), Anti-S5a (CPTC-PSMD4 hybridoma obtained from the Developmental Studies Hybridoma Bank, developed under the auspices of the NICHD and maintained by the University of Iowa, Department of Biology, Iowa City, IA, USA), anti-T7 (ICL RT745AZ) and anti-PSMA1 (ABR PAI-963 or Mab. 2–17 to human PSMA1, a kind gift from Keiji Tanaka, Tokyo, Japan).

### 2.5. Recombinant AIRAP Purification and PAO Binding Assay

AIRAP cDNA was subcloned into pET30a (EMD Bioscience, San-Diego, CA, USA) to yield a C-terminus His tag. Overnight auto-induction in BL21 bacteria [31] was performed at 18 °C and Ni-NTA (GE Healthcare, Chicago, IL, USA) purifications were performed following the manufacturer’s procedure. Prior to protein elution from the matrix, an additional wash with 10 mM THP was performed in order to maximize protein reduction. Eluted protein was buffer exchanged for NH buffer (50 mM Hepes-7.5, 150 mM Nacl) using a PD Sephadex G25 column (GE Healthcare, Chicago, IL, USA).

PAO beads were purchased from Toronto research chemicals (cat#A622500, Toronto, ON, Canada), dissolved in methanol and conjugated to affigel (BioRad, Hercules, CA, USA) by mixing at R.T. for 2 h. Aminoethanol was thereafter added to block any remaining active sites, and the sample was extensively washed with methanol followed by PBS washes. Control beads (assigned as control in the relevant figures) were prepared by aminoethanol blocked beads. Recombinant AIRAP in the presence or absence of the indicated compounds (Figure 1) was mixed with the PAO beads at R.T., followed by extensive washes with PBS and resuspension in Laemmli buffer. Bound material was loaded onto an SDS-PAGE and AIRAP content was evaluated by immunoblot.

### 2.6. ICP-OES and MST Measurements

For ICP-OES measurements, samples of Ni-NTA bound recombinant AIRAP (25 μM) were prepared with arsenite and zinc pre-wash at the indicated concentrations (0.1–10 mM) at R.T. for 10 min. Samples were washed with HN buffer, eluted with 400 μL elution buffer (20 mM phosphate buffer pH 7.5, 150 mM NaCl, 400 mM imidazole) and diluted with 4 mL DDW and 3% nitric acid. Zinc and arsenite content were measured by Optima 3000 (PerkinElmer, Waltham, MA, USA) calibrated with Merck standards. For MST measurements, recombinant AIRAP (1 μM) was incubated with a series of 16 dilutions of arsenite (range 0.6 nM–20 μM). AIRAP samples gave a significantly higher intrinsic tryptophane fluorescence signal than the HNT buffer (HN buffer with 1 μM Tween-20). Samples were loaded into glass capillaries and measured by Monolith LabelFree (Nanotemper GmbH, Munich, Germany).

## 3. Results

### 3.1. Recombinant AIRAP Binding to Arsenite

Arsenite-induced protein misfolding occurs through binding to vicinal cysteine residues [32]. AIRAP is a short polypeptide highly enriched with cysteine residues that coordinate through two AN1 type zinc fingers the binding of zinc residues. To evaluate the possibility of the direct binding of AIRAP to arsenite, we utilized an immobilized bivalent form of arsenite (phenylarsine oxide (PAO)) previously demonstrated to bind arsenite-binding proteins [33].

Recombinant AIRAP was incubated with PAO beads, unbound material was extensively washed, and AIRAP immunoblot was performed to evaluate the bound content (Figure 1). Several controls were performed to evaluate the nature of this binding, and the protein configuration was evaluated by preincubating the recombinant protein with guanidine. The modification of AIRAP cysteine content was evaluated by preincubation with *N*-ethylmaleamide (NEM) or the reducing agent THP. Finally, a trivalent arsenite (As^3+^) and a bivalent form of arsenite were used to evaluate their ability to outcompete the immobilized PAO binding to AIRAP. As evident from the results in Figure 1A, the binding of AIRAP was highly dependent on the protein’s native structure (guanidine) and cysteine configuration (NEM and THP). Furthermore, AIRAP binding efficiently competed with a trivalent form of arsenite but not with a bivalent form (arsenite and PAO). To further investigate the specificity of AIRAP’s ability to bind arsenite and the effect of different metals on AIRAP’s ability to bind the PAO matrix, we incubated recombinant AIRAP with PAO matrix followed by incubation with various metals. As indicated in Figure 1B, metals such as As^3+^ and Fe^3+^ reduced AIRAP’s binding to the PAO matrix (Figure 1B, lanes 3 and 7). These results imply a selective metal binding by AIRAP to the PAO matrix.

As AIRAP is a zinc-binding protein, zinc may serve as a primary structural enhancer to enable the optimal positioning of cysteine residues to bind arsenite. This notion is further supported by our results with the PAO matrix (Figure 1B). To directly measure the protein’s capacity to bind arsenite in a zinc-dependent manner, we measured arsenite binding to AIRAP by inductively coupled plasma optical emission spectroscopy (ICP-OES). Purified AIRAP was incubated with increasing concentrations of arsenite in the presence or absence of zinc, and arsenite content was evaluated. As seen in Figure 2A, AIRAP binding to arsenite was greatly enhanced when first incubated with zinc (marked Zn^+^). These results coincided with the indicated increase in AIRAP binding to the PAO matrix upon zinc incubation (Figure 1B). Analytical evaluation of this interaction was performed to determine the Kd value of AIRAP to arsenite using a micro-scale thermophoresis (MST) methodology [34]. A Kd of 7 nM ± 0.87 nM was determined (Figure 2B), demonstrating the high affinity of AIRAP toward arsenite.

### 3.2. AIRAP Binding to Arsenite In Vivo

After examining the in vitro binding of AIRAP to arsenite, we addressed the ability of endogenous AIRAP to bind arsenite. To this end, we used the cell lysate as a source for AIRAP. Upon endogenous expression of AIRAP (induced by arsenite treatment), AIRAP could not be detected on the PAO matrix from total lysate (Figure 3, lanes 2 and 3) but could be detected from purified proteasomes (Figure 3, lanes 5 and 6). It is important to note that the interaction was stable even after a high-salt wash, as would be expected from a metal-coordinated binding to the PAO matrix (lane 6). In addition, although catalytic particle subunit PSMA1 was present in the total lysate and the proteasome fractions (Figure 3 lane 4), it was barely detectable in the PAO matrix-purified fractions; thus, PSMA1 served as a specificity control for the PAO matrix binding. We reasoned that the lack of AIRAP binding to the PAO matrix in the total lysate may have been due to the existence of other proteins that competed towards the PAO matrix binding, or it may have indicated a structural change that AIRAP may undergo upon proteasome binding. This transition may be intended towards optimal binding to arsenite only upon proteasome binding.

### 3.3. Endogenous Rpn11/PSMD14 Binds PAO Matrix More Strongly Than Other Proteasome Subunits

After establishing the ability of AIRAP to bind arsenite, we sought to understand the extent to which arsenite-binding proteins bind the proteasome and how this function may relieve UPS impairment upon arsenite exposure. We reasoned that the 26S proteasome has one metal-binding protein, Psmd14/Rpn11, that functions as a proteasomal deubiquitination enzyme [35,36]. Possible impairment of Psmd14 activity may occur if arsenite binding perturbs metalloprotease activity, as previously reported for Rpn11 metal selectivity [37]. This, in turn, will lead to polyubiquitin accumulation and impaired turnover of UPS substrates. We therefore sought to determine whether Psmd14 can bind the PAO matrix. Affinity-purified soluble proteasomes were mixed with the PAO matrix and the Psmd14, S5a (an additional 19S subunit) and Psma1 (20S subunit) content was evaluated by immunoblot. As seen in Figure 4, Psmd14 was enriched on the PAO matrix. It is important to note that other proteasome subunits, while present, were not enriched as significantly as Psmd14 on the PAO matrix (compare PAO matrix lane and input lane for all evaluated proteasome subunits). These results are in line with Psmd14 being the proteasomal target for AIRAP binding.

### 3.4. Arsenite Regulation of Psmd14 Activity via MPN/JAMM Motif

Psmd14 is a zinc-binding protein that contains an N-terminal MPN/JAMM motif, with two reactive histidines and one aspartic acid, which coordinate a zinc ion and catalyze deubiquitination [35,36,38]. In order to evaluate Psmd14′s ability to bind the PAO matrix through its MPN/JAMM motif, we incorporated a mutant version of Psmd14 that lacked the two reactive histidines of the MPN/JAMM motif. Purified proteasomes acquired from transfected cells were examined towards Psmd14′s ability to bind the PAO matrix. While both endogenous and exogenous Psmd14 were present at purified proteasome fractions (Figure 5, lanes 1,3 and 5), only the WT and not the mutant Psmd14 was able to bind the PAO matrix (Figure 5, compare lanes 4 and 6). These results suggest that Psmd14 can bind arsenite, a feature that requires a functional MPN/JAMM motif.

Previous reports demonstrated arsenite-dependent induction in the cellular content of polyubiquitinated proteins [24,26]. AIRAP’s ability to bind metals such as arsenite and its role as an adaptor of the proteasome might be connected to Psmd14′s ability to bind arsenite (Figure 5). We hypothesized that arsenite is a competitor of zinc in Psmd14’s active site. In order to evaluate the toxic effect of arsenite on Psmd14′s activity, we sought to establish a Psmd14 activity assay. Since mammalian proteasomes contain three DUB subunits (Psmd14, Usp14 and Uch37), we first purified proteasomes that contained only the Psmd14 DUB (19SΔUU proteasomes, Figure 6A) by using high-salt washes [39]. Next, we used an in vitro degradation assay in which purified proteasomes that contained only the Psmd14 DUB (19SΔUU proteasomes) processed a polyubiquitinated Sic1 ([30,36]; Figure 6B). While the 19SΔUU had reduced DUB activity when compared to the 19S particle (data not shown), the deubiquitination of the Sic1 substrate was evident and enabled us to evaluate directly Psmd14 DUB activity in respect to arsenite and AIRAP modulation without the involvement of Usp14 and Uch37. We examined the effect of arsenite on Psmd14 activity by treating purified 19SΔUU proteasomes with arsenite and noted the inhibition of polyubiquitinated Sic1 clearance upon arsenite treatment (Figure 6B, compare lanes 3–6 and lanes 7–10). Next, we evaluated the ability of AIRAP to restore DUB activity in vitro by adding recombinant AIRAP to the arsenite-treated proteasomes. As shown in lanes 11–14, AIRAP restored Psmd14 DUB activity upon arsenite-inflicted inhibition. This result implies a role for AIRAP in proteasomal activity that directly involves Psmd14 DUB activity.

## 4. Discussion

Direct perturbation of proteasome activity gives rise to polyubiquitin accumulation, attesting to the continuous degradation of proteins in the cell by the proteasome. However, protein misfolding conditions that increase UPS substrate levels cause only minor changes in polyubiquitin levels in the cell ([26], Figure 1 therein). This is attributed to many cellular responses to protein misfolding that, in concert, are able to maintain a continuous flux of UPS substrates. In this aspect, arsenite differs from other protein misfolding conditions as it induces high accumulation in polyubiquitin levels in the cell. Our results presented herein demonstrate the direct binding of arsenite to the proteasomal metalloprotease Psmd14. This result reveals arsenite as a unique protein misfolding agent that impacts directly on the UPS. This inhibitory effect of Psmd14 activity is unique as various other metals were not demonstrated to reduce in vitro Psmd14/Rpn11 DUB activity [37]. We do not believe that Psmd14 is unique in respect to other metal-binding proteins; in fact, our PAO binding of AIRAP or Psmd14 was not obtained from the total cell extract, as we posited that many metal-binding proteins with higher affinities were competing for this binding. This may include other JAMM metalloproteases such as Csn5 signalosome. However, in a cellular context, many proteins containing vicinal thiols maybe misfolded by arsenite, ubiquitinated and subjected to proteasomal degradation. Post-recognition and during their unfolding on the proteasome, the metalloprotease Psmd14 maybe exposed to locally high concentrations of arsenite and/or the unfolding of the arsenylated UPS substrate may reduce the substrates’ affinity to arsenite, thereby favoring the arsenylation of Psmd14, which is located nearby [40,41]. Structural data of Rpn11 indicate a catalytic role rather than a structural role for zinc in Rpn11, thereby supporting the notion that Psmd14 will retain its metal-binding capacity to support metal exchange [42]. Based on this, JAMM motif mutations that impair zinc binding (Psmd14 D126A) would enable us to further investigate the role of arsenite in proteasomal DUB inhibition. Establishment of *PSMD14* D126A knock-in cells is a desirable perspective for such investigations that would enable us to address the AIRAP–PSMD14 relation in the context of an intact 19S particle. In addition to the above, the selective effect of metal substitution on deubiquitination activity [37] further supports the feasibility of metal exchange and the specificity of arsenite rather than other metals in inhibiting proteasome activity. As mentioned above, the unique ability of arsenite to inhibit UPS activity can now be explained by Psmd14 inhibition.

The AIRAP gene induction program is tuned to proteasomal dysfunction via the NRF and HSF pathways [24,25] and not specific to arsenite. Binding of AIRAP to the proteasome in the vicinity of Psmd14 [24] would enable a molecular relay of arsenite atoms from Psmd14 to AIRAP, thereby enabling revival of the proteasomal flux. After the clearance of arsenite atoms from Psmd14, we speculate that AIRAP may serve as a donor for a second relay event of a zinc atom back to Psmd14, thereby completing a zinc for arsenite exchange between Psmd14 and AIRAP. This scenario is in line with the reduced clearance of polyubiquitin accumulation observed in the AIRAP^-/-^ cells upon acute exposure to arsenite and to the elevated polyubiquitin found in proteasomes acquired from AIRAP^-/-^ cells [24]. Establishing this hypothesis will require further work in respect to Psmd14 inhibition by arsenite and structural data of an AIRAP–26S proteasome complex.

## Figures and Tables

**Figure 1 biomolecules-11-01317-f001:**
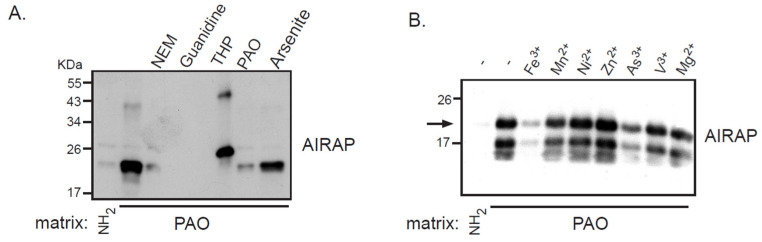
Recombinant AIRAP binding to a PAO matrix. (**A**) Recombinant mouse AIRAP was expressed as a C-terminus 6×His fusion protein and Ni-NTA affinity purified. Purified material was incubated with a control (NH_2_) or amino-phenylarsine (PAO) immobilized Sepharose 4B matrix in the presence of the cysteine alkylating *N*-ethylmaleamide (NEM), denaturing conditions (6M guanidine–HCl), reducing conditions (THP), sodium arsenite (As) or soluble amino-phenylarsine (PAO). Immobilized material was extensively washed, eluted by boiling with 2% SDS and subjected to SDS-PAGE and subsequent immunobloting to evaluate AIRAP content. (**B**) The impact of metals on AIRAP binding to a PAO matrix was performed as in 1A by performing the binding in the presence of the indicated metals. The arrow indicates the full-length recombinant product of AIRAP.

**Figure 2 biomolecules-11-01317-f002:**
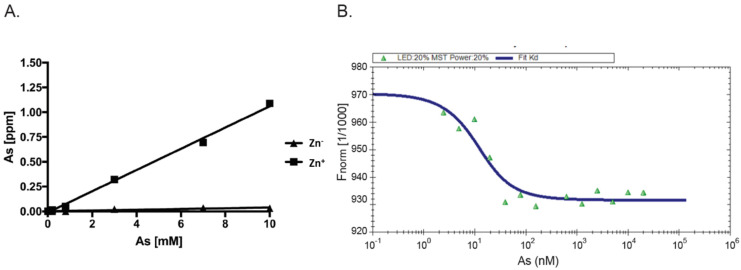
Dependence on zinc and MST measured affinity. (**A**) The presence of arsenite bound to AIRAP was determined by ICP-OES. Recombinant AIRAP was incubated in the presence (Zn^+^) or absence (Zn^−^) of zinc, prior to arsenite exposure. Bound material was extensively washed and arsenite content determined. (**B**) Recombinant AIRAP (1 μM) was incubated with a range of arsenite dilutions as indicated, and intrinsic fluorescence was measured by MST. Analysis revealed Kd values of 7 nM ± 0.87 nM.

**Figure 3 biomolecules-11-01317-f003:**
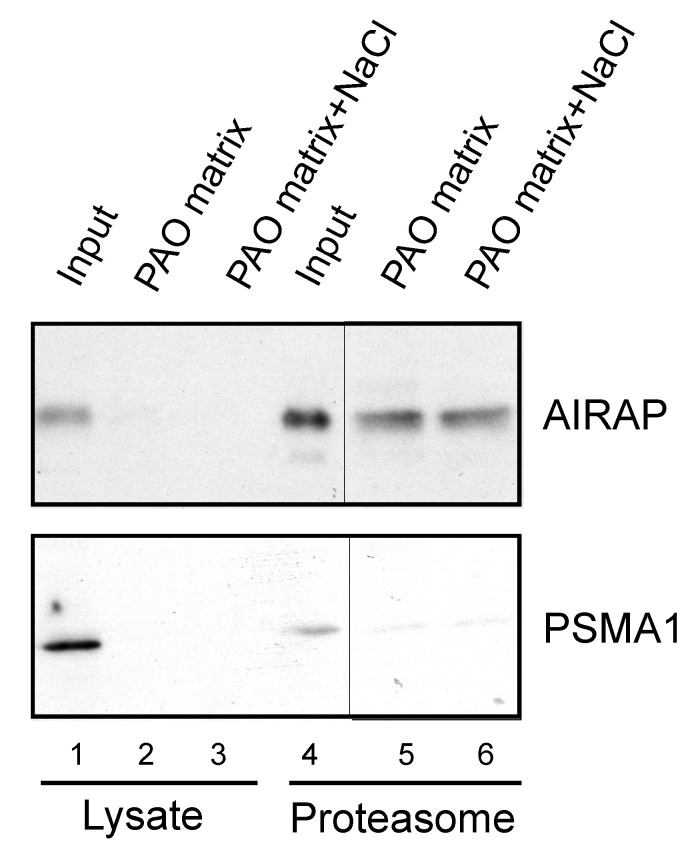
Endogenous proteasomal AIRAP binds arsenite. Cell lysates acquired from arsenite-treated cells were incubated with a PAO matrix. AIRAP and 20S proteasomal content (PSMA1) were evaluated by immunoblot (lanes 1–3). Alternatively, PAO matrix binding was performed using soluble proteasomes purified from arsenite-treated cells (lanes 4–6). Where indicated, PAO matrix was washed with high salt to evaluate impact on PAO binding (lanes 3 and 6).

**Figure 4 biomolecules-11-01317-f004:**
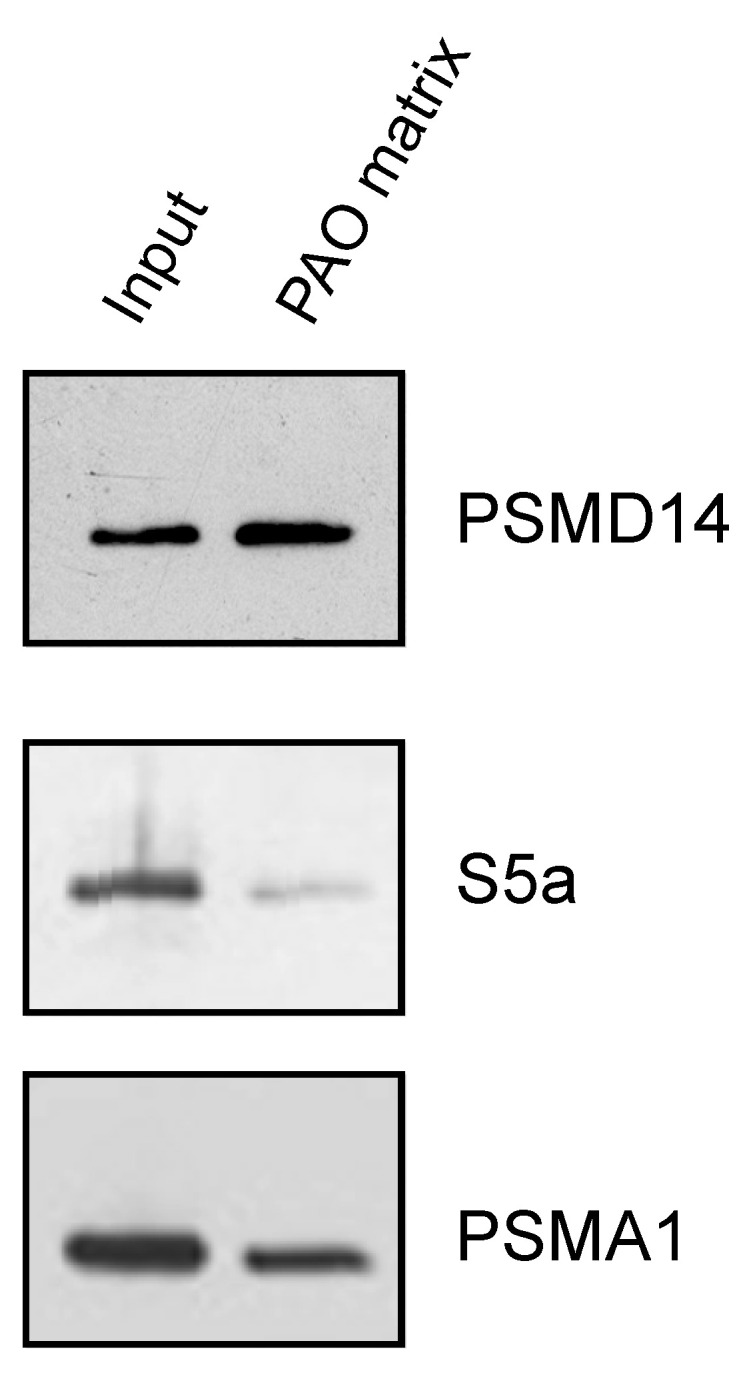
PSMD14 binding to PAO. Soluble proteasomes were incubated with PAO matrix, extensively washed and PSMD14, 19S (S5a), and 20S (PSMA1) content revealed by immunoblot.

**Figure 5 biomolecules-11-01317-f005:**
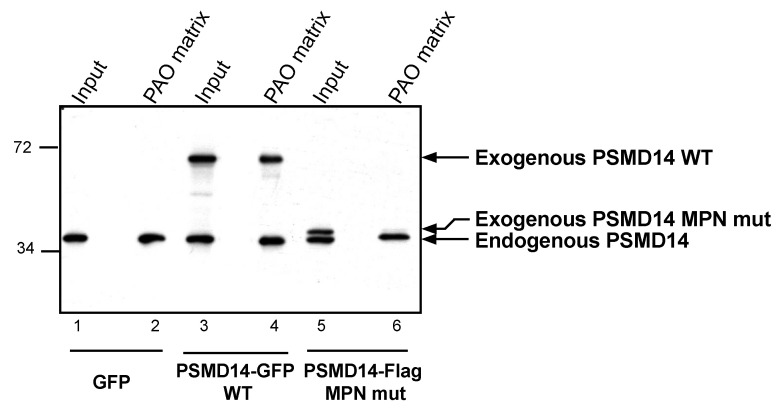
MPN domain mediates PSMD14 binding to PAO. PSMD14 WT or MPN mutant was evaluated towards its ability to bind a PAO matrix. Proteasomes were purified from cells expressing GFP (serving as a control), PSMD14-GFP (WT) or PSMD14-Flag (MPN mutant). Proteasomal content (input) reveals no requirement for a functional MPN domain for proteasomal binding (lanes 3 and 5). However, PSMD14 arsenite binding (PAO matrix) is detected only in the PSMD14 WT and not MPN mutant (lanes 4 vs. 6).

**Figure 6 biomolecules-11-01317-f006:**
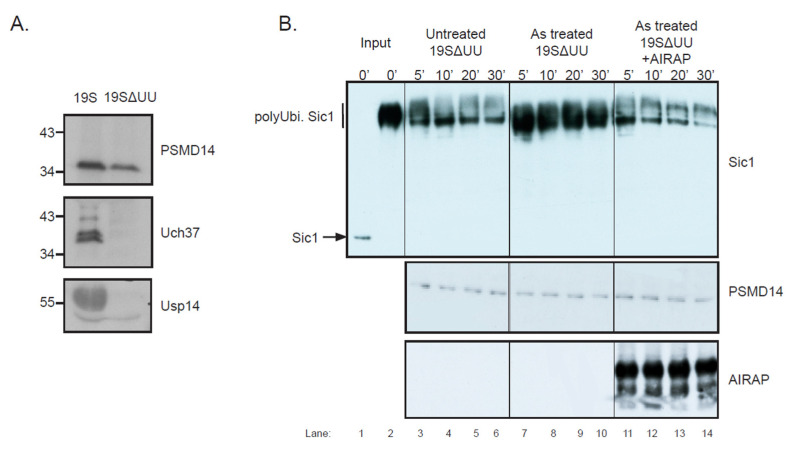
Proteasomal PSMD14 activity assay. (**A**) 19SΔUU proteasomes containing the PSMD14/Rpn11 DUB do not contain Usp14/Ubp6 or Uch37, while WT proteasomes contain all three DUBs. (**B**) Polyubiquitinated Sic1 was mixed with 19SΔUU proteasomes and samples were retrieved at the indicated time points. Proteasomal PSMD14, AIRAP and Sic1 content was evaluated by immunoblot. Where indicated, arsenite was added to the reaction prior to substrate addition without (lanes 7–10) or with (lanes 11–14) recombinant AIRAP (as indicated). Lanes 1 and 2 (input) are recombinant Sic1 and in vitro polyubiquitinated Sic1 without proteasomes.

## Data Availability

Not applicable.

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
