# Peer review of "An Arsenite Relay between PSMD14 and AIRAP Enables Revival of Proteasomal DUB Activity"

_biomolecules, 2021, doi:10.3390/biom11091317_

Round 1

Reviewer 1 Report

This is an interesting paper. I only have a few minor questions/points.

  1. The conclusion that Zn2+ increases binding in Fig. 1B is questionable. Based on one slot on a WB without loading control (and this would not have been possible). The experiment is fine but the text could be modified ("while others such as Zn2+ increase it (figure 1B, lane 3, 7 compared with lane 6)". The "others" include Ni2+? The authors could include three independent experiments and a t-test. However, given the remaining data I don´t think this is necessary (I would recommend just concluding that As3+ and Fe3+ decreases binding).
  2. Fig. 6 is difficult for the reader to interpret. The substrate is not completely deubiquitinated (I would have expected this (?), is there trimming DUB activity remaining in the preps?). The degree of deubiquitination is difficult to evaluate on the gels used (a lower percentage gel would resolve the molecules better). At least the authors should help the reader by discussing the results in more detail.
  3. The Discussion is only one paragraph and rather speculative. It is a complex problem and I understand the authors' problem in writing the section. But I recommend a more thorough Discussion, perhaps one section with Rpn11 with a conclusion and a second with AIRAP? 

Author Response

1-The conclusion that Zn2+ increases binding in Fig. 1B is questionable. Based on one slot on a WB without loading control (and this would not have been possible). The experiment is fine but the text could be modified ("while others such as Zn2+ increase it (figure 1B, lane 3, 7 compared with lane 6)". The "others" include Ni2+? The authors could include three independent experiments and a t-test. However, given the remaining data I don´t think this is necessary (I would recommend just concluding that As3+ and Fe3+ decreases binding).

In the revised manuscript we have changed text as suggested.

2-Fig. 6 is difficult for the reader to interpret. The substrate is not completely deubiquitinated (I would have expected this (?), is there trimming DUB activity remaining in the preps?). The degree of deubiquitination is difficult to evaluate on the gels used (a lower percentage gel would resolve the molecules better). At least the authors should help the reader by discussing the results in more detail.

The 19DUU proteasomes have reduced DUB activity as they lack the Usp14 and Uch37 DUBs. This is evident from comparative deubiquitination reactions we have performed with the poly-ubiquitinated Sic1 (these experiments were performed in respect to characterizing Rpn11 activity unrelated to AIRAP). This would explain the lack of complete deubiquitination of the substrate. As suggested, in the revised manuscript we have extended the description of the assay and discuss the importance of using the 19DUU to better clarify the points raised by the reviewer.

3-The Discussion is only one paragraph and rather speculative. It is a complex problem and I understand the authors' problem in writing the section. But I recommend a more thorough Discussion, perhaps one section with Rpn11 with a conclusion and a second with AIRAP? 

To the reviewers request we have extended the discussion section to better explain the complex issues

Reviewer 2 Report

This study aims to increase our understanding of interplay between AIRAP and the proteasome during arsenite stress. AIRAP has previously been demonstrated to be induced by arsenite stress (and other types of stress that lead to HSF1 activation) as well as associating with the proteasome. Here, the authors investigate the metal binding, proteasome binding and regulation of PSMD14 deubiquitinase activity by AIRAP. Overall, the paper is nicely written, I especially applaud the methods section for its degree of detail.  For the most part conclusions are supported by the data, but there are situations in which I think additional discussion of the data. Below I suggest minor changes/additions that I think would improve the manuscript

  1. As a reader, it would have been helpful to me if the AIRAP-related part of the introduction had been more detailed. If it is a concern that the introduction is too long, I would suggest making the general proteasome part of the introduction shorter.

  1. The legend to figure 1 states that 1B was performed as in 1A but in the presence of the indicated metals. Could the authors explain why there is only one band in 1A but multiple in 1B? Adding arrows to indicate the specific AIRAP band would also be helpful. 

  1. Have the proteasomes in figure 3 and 4 been purified the same way? In figure 4 the binding of PSMA1 to the PAO matrix is obvious but it is undetectable in figure 3. Please clarify this discrepancy.

  1. Figure 4 and 5: Would any JAMM/MPN motif containing protein bind to the PAO matrix? I apologize if this is a naïve question

  1. Figure 6: I think the authors need to discuss if the ability of AIRAP to relieve PSMD14 inhibition is relevant in a physiological context in which Uch37 and Usp14 are also present? Do the authors think accumulation of polyubiquinated species in AIRAP (-/-) cells can be accounted for solely by the lack of the mechanism which they propose? I would like to see a side by side comparison between 19S and 19SΔUU proteasomes in this assay. If the effect of AIRAP is only visible in an 19SΔUU context, is it then relevant in a cellular context? 

  1. “Our results illustrate a cellular mechanism that has evolved to sense proteasomal inhibition and restore PSMD14/Rpn11 activity by a metal relay between arsenylated PSMD14/Rpn11 and AIRAP” I think this final sentence of the abstract should be toned down a notch, since the presented data is exclusively in vitro. Also, I don’t find sufficient experimental evidence to claim that a metal relay is happening. It is possible, but not demonstrated in the presented data. 

Author Response

1-As a reader, it would have been helpful to me if the AIRAP-related part of the introduction had been more detailed. If it is a concern that the introduction is too long, I would suggest making the general proteasome part of the introduction shorter.

In the revised manuscript we have extended this introduction part of AIRAP in more detail as requested by the reviewer.

2-The legend to figure 1 states that 1B was performed as in 1A but in the presence of the indicated metals. Could the authors explain why there is only one band in 1A but multiple in 1B? Adding arrows to indicate the specific AIRAP band would also be helpful. 

The LMW band is a partial degradation product of recombinant AIRAP and we have added an arrow to indicate the full-length species of AIRAP. In is noteworthy that both species behave identically. This is now stated in the figure legend of the revised manuscript. The differences between figure 1A and 1B are due to variations in preparations of recombinant AIRAP obtained during different preparations (using the same protocols) and as mentioned represent a  partial degradation product of AIRAP.

3-Have the proteasomes in figure 3 and 4 been purified the same way? In figure 4 the binding of PSMA1 to the PAO matrix is obvious but it is undetectable in figure 3. Please clarify this discrepancy.

The differences that seem regarding PSMA1 binding are actually a qualitative difference and exposures. Note that faint binding of PSMA1 is seen in Figure 3 and much weaker than the AIRAP immunoblot. The weak binding of PSMA1 can easily be part of a AIRAP-proteasome complex. The proteasomes in figure 3 and figure 4 were prepared using the mAb. To PSMA1 or S5a as described in the material and method section. In figure 3 that addresses the binding of endogenous AIRAP to the PAO matrix, the purifications were performed from MEF cells as the AIRAP antisera we obtain detects mouse isoform of AIRAP much better than the human isoform. Since the mAb. to PSMA1 does not detect the mouse isoform, we purified the proteasomes using th S5a mAb. As such the retained 20S particle ratio is lower and was detected with a polyclonal antisera that does recognized the mouse isoform, and was detected very weakly. In Figure 4 we addressed the binding of Psmd14 to the PAO matrix and used human 293 cells and proteasomes were purified with the PSMA1 Mab. This is why in Figure 4 PSMA1 is detected much stronger as it was the 20S entity used for purifications. While the signals of proteasome subunits bound to the PAO matrix are stronger in figure 4 than that compared to figure 3 (explained above), only PSMD14 shows enrichment on the PAO matrix while S5a and PSMA1 show reduction. Retaining quantities maybe explained as in-direct binding of the whole 26S proteasome.

4-Figure 4 and 5: Would any JAMM/MPN motif containing protein bind to the PAO matrix? I apologize if this is a naïve question

This is a very good point, especially in respect to other JAMM motif DUBs such as the Csn5. This point is probably connected to the inability to bind PSMD14 and AIRAP to bind PAO directly from cell lysates as other metal binding proteins are probably competing for the matrix binding. In the revised manuscript we have added in the discussion section a note that address this point.

5-Figure 6: I think the authors need to discuss if the ability of AIRAP to relieve PSMD14 inhibition is relevant in a physiological context in which Uch37 and Usp14 are also present? Do the authors think accumulation of polyubiquitinated species in AIRAP (-/-) cells can be accounted for solely by the lack of the mechanism which they propose? I would like to see a side by side comparison between 19S and 19SΔUU proteasomes in this assay. If the effect of AIRAP is only visible in an 19SΔUU context, is it then relevant in a cellular context? 

The 19DUU proteasomes have reduced DUB activity as they lack the Usp14 and Uch37 DUBs. This is evident from comparative deubiquitination reactions we have performed (data not shown) with the poly-ubiquitinated Sic1 (these experiments were performed in respect to characterizing Rpn11 activity unrelated to AIRAP). Experiments suited to better address AIRAP relation with PSMD14 in context of a 19S containing Usp14 and Uch37 should include generation of PSMD14 D126A knock-in cells. In this scenario, we should be able to show the specific effect of AIRAP on PSMD14 activity in context of a relevant 19S particle that contains all DUBs. This tool has not yet been created and is a long term goal for our future investigation. In the revised manuscript we have addressed this point and have added in the discussion section a note concerning this point.

As to addressing the accumulation of poly-ubiquitinated species in the AIRAP -/- cells, we anticipate that a PSMD14 D126A mutant in the background of AIRAP -/- cells could help address this point assuming basal poly-ubiquitinated levels of PSMD14 D126A cells are not too high.

6-Our results illustrate a cellular mechanism that has evolved to sense proteasomal inhibition and restore PSMD14/Rpn11 activity by a metal relay between arsenylated PSMD14/Rpn11 and AIRAP” I think this final sentence of the abstract should be toned down a notch, since the presented data is exclusively in vitro. Also, I don’t find sufficient experimental evidence to claim that a metal relay is happening. It is possible, but not demonstrated in the presented data. 

In the revised manuscript we have revised our statement, tuned down to better suit the presented results and present a relay mechanism only as a possibility.

Round 2

Reviewer 2 Report

I appreciate the authors' clarifying responses to the questions I raised.